



# A comparison of dynamic inflow models for the Blade Element Momentum method

Simone Mancini[1], Koen Boorsma[1], Gerard Schepers[1], and Feike Savenije[1]

[1]TNO Wind Energy, Westerduinweg 3, 1755LE Petten, Netherlands

**Correspondence:** Simone Mancini (simone.mancini@tno.nl)

**Abstract.**

With the increase in rotor sizes, the implementation of innovative pitch control strategies, and the first floating solutions entering the market, the importance of unsteady aerodynamic phenomena in the operation of modern offshore wind turbines has increased significantly. Including aerodynamic unsteadiness in Blade Element Momentum (BEM) methods used to simulate wind turbine design envelopes requires specific sub-models. One of them is the dynamic inflow model, which attempts to reproduce the effects of the unsteady wake evolution on the rotor plane induction. Although several models have been proposed, the lack of a consistent and comprehensive comparison makes their relative performance in the simulation of large rotors still uncertain. More importantly, different dynamic inflow model predictions have never been compared for a standard fatigue load case and thus it is not clear what is their impact on the design loads estimated with BEM. The present study contributes to filling these gaps by implementing all the main dynamic inflow models in a single solver and comparing their relative performance on a 220m diameter offshore rotor design. Results are compared for simple prescribed blade pitch time histories in uniform inflow conditions first, verifying the predictions against a high-fidelity free vortex wake model and showing the benefit of new two-constant models. Then the effect of shed vorticity is investigated in detail revealing its major contribution to the observed differences between BEM and free vortex results. Finally, the simulation of a standard fatigue load case prescribing the same blade pitch and rotor speed time histories reveals that including a dynamic inflow model in BEM tends to increase the fatigue loads, while the relative differences among the models are limited.

## 1 Introduction

Besides airfoil unsteady effects, which have been long studied by the aerospace community, wind turbines experience unsteadiness also at a rotor level due to the dynamics of their wakes. Wake-related unsteady effects are typically referred to as dynamic inflow, dynamic wake, or dynamic induction. The underlying phenomenon is the time lag with which the induced velocity field adapts to changes in rotor plane conditions, e.g. because of sudden controller actions or wind gusts. Such a lag depicts the time needed for the old wake vorticity, shed before the change, to convect far enough downstream not to influence the induction on the rotor plane anymore.

Including dynamic inflow effects in Blade Element Momentum (BEM) methods requires the implementation of some engineering sub-models (Schepers, 2012). The first two dynamic inflow models for Horizontal Axis Wind Turbines (HAWT) date



back to the 90s (Snel and Schepers, 1994), and they are still implemented in some state-of-the-art BEM codes after more than two decades. The first was proposed by the Energieonderzoek Centrum Nederland (ECN, now part of TNO) and it consisted of the addition of a first-order term to the axial momentum equation featuring a radially dependent time constant derived analytically from a cylindrical wake assumption. The other was developed by Øye (1990), who used two first-order filters to correct the quasi-steady induction value found by the momentum theory. The parameters were calibrated on a simple vortex ring wake model of a uniformly loaded Actuator Disk (AD). Each filter had an associated time constant: a fast one with a strong radial dependency to represent near wake effects; and a slow one, independent of the radial location to account for the far wake. The Øye model is currently used in the BEM solvers of OpenFAST (NREL) and Bladed (Beardsell et al., 2016) while the ECN model is till implemented in AeroModule (Boorsma et al., 2011), for example.

As new experimental data became available to the wind energy community, several numerical-experimental validation campaigns were conducted (e.g., Schepers, 2007; Schepers et al., 2014) refuting the radial variation of the ECN model time constant and concluding that two-constant models should be preferred (Sørensen and Madsen, 2006; Schepers et al., 2018b; Pirrung and Madsen, 2018). Building on this insight, Yu et al. (2019) exploited the theory of linear systems to derive a simple expression for the axial induced velocity of a uniformly loaded actuator disc undergoing step variations of the thrust coefficient ($C_T$). The model relies on two indicial functions calibrated with both linear and non-linear vortex ring models. A verification against AD-CFD results showed that the new model (especially the "TUD-VR" version calibrated on a non-linear wake model) offered better predictions than Øye and ECN dynamic inflow models. Despite its promising results, to the best of the authors' knowledge, this model has never been implemented in a state-of-the-art BEM code and therefore its performance on real rotors has not been verified yet. Another model was recently presented by Madsen et al. (2020), from the Danmarks Tekniske Universitet (DTU), and implemented in the BEM solver of HAWC2 (Horizontal Axis Wind turbine simulation Code, 2007). Like the Øye model, the DTU model relies as well on two first-order filters to correct the quasi-steady induction, but it uses new filter functions that were calibrated on step $C_T$ variations of a uniformly loaded AD modelled with CFD. The latest development on dynamic inflow models for BEM is the work of Berger et al. (2022) who studied dynamic wake effects triggered by wind gusts on the MoWiTO turbine (a 1.8m diameter scaled version of the NREL 5MW), exploiting the active grid installed in the wind tunnel of ForWind - University of Oldenburg (Neuhaus et al., 2021). Based on the comparison of BEM results with experimental measurements the authors suggested an improvement of the Øye model formulation to account for gust-driven rotor unsteadiness.

Trying to identify the reasons for the overestimation of fatigue loads with BEM codes observed in AVATAR (2018c), the project VortexLoads (2020) revealed that in non-uniform inflow conditions the dynamic wake sub-model and its implementation may result in artificial damping of the induced velocity variations leading to larger aerodynamic loads. Similar evidence was also found by Perez-Becker et al. (2020). Since implementation choices may vary greatly among different BEM codes (Schepers et al., 2018a; Madsen et al., 2020), it is still unclear whether it is the specific dynamic inflow model or the way it is implemented in the solver that matters most. One reason is the lack of benchmark studies comparing all available dynamic inflow models on real rotor cases. A fair comparison requires the dynamic inflow model to be the only variable changing indeed, and this is hardly possible when results from different codes are compared. An exception is the work of Berger et al. (2020), in





which the predictions of different dynamic inflow models (implemented in a single BEM code) were compared against wind tunnel measurements and results of higher fidelity tools. Fast blade pitch steps on the 1.8m diameter MoWiTO turbine were considered as a test case. The comparison confirmed that two-constant models provide better results than the ECN model, and quite similar predictions between DTU and Øye models were found. The TUD-VR model was not included, however.

An accurate modelling of dynamic inflow effects is also important for controller design (van Engelen and van der Hooft, 2004), especially when it comes to innovative Individual Pitch Control (IPC) strategies where the blade is pitched at relatively high frequencies. Moreover, the characteristic time scale for rotor unsteadiness is proportional to the turbine diameter ($\tau = \frac{D}{V_0}$, with $\tau$ being the time scale, $V_0$ the free-stream wind velocity, and $D$ the diameter) hence dynamic wake effects in large rotors can be expected already at lower frequencies. Finally, dynamic inflow research is also very active on floating wind turbines
(e.g. Mancini et al., 2020; Ferreira et al., 2021) where unsteadiness is triggered by floater motions.

    The dynamic inflow modelling activities carried out in this work are grounded in this context. Two main targets were set for this research:

- – Implement the main dynamic inflow models in a single BEM solver to isolate differences and assess their relative performance compared to FVW predictions for a large offshore rotor.

- – Understand the impact that the different dynamic inflow models have on a fatigue Design Load Case calculation (DLC).

    This paper presents the main results of the study, comparing the different dynamic inflow model predictions and providing new insight into their differences as well as on the effect of shed vortices on the unsteady aerodynamic response of a large offshore rotor. Finally, the practical impact of BEM dynamic inflow models on a fatigue load case is assessed.

    The article is structured as follows: the dynamic inflow models and their implementation are described in Sect. 2; the
numerical results are discussed in Sect. 3, addressing blade pitch steps (Sect. 3.1), sinusoidal pitch variations (Sect. 3.2), and the impact on BEM predictions for the standard DLC 1.2 (Sect. 3.3); conclusions and suggestions for future research are addressed in Sect. 4; the expressions used to implement the new models are reported in Appendix A; additional results from DLC 1.2 are shown in Appendix B.

## 2   Numerical models

This section introduces the numerical models that have been used to obtain the results presented in Sect. 3. The AeroModule library and its aerodynamic solvers are introduced first (Sect. 2.1), and then the different dynamic inflow models for BEM are presented (Sect. 2.2) along with their implementation in AeroModule (Sect. 2.3). Finally, a brief description of the wind turbine rotor considered and the main simulation settings are given in Sect. 2.4.

### 2.1   TNO AeroModule

The AeroModule (AM, Boorsma et al., 2011) is a state-of-the-art wind turbine aerodynamic library that includes both a BEM solver and a Free Vortex Wake (FVW) model called AWSM (van Garrel, 2003). AM can be either coupled to a structural code





to perform aeroelastic load case calculations or used as a standalone tool for simple aerodynamic design iterations featuring a rigid turbine model. Thanks to the source code modularity, BEM and AWSM share multiple routines that guarantee fully consistent inflow modelling and airfoil aerodynamic properties (the same polar look-up tables are used) including rotational

effects and dynamic stall models. This allows the user to switch between the two solvers from the same input file, and hence facilitates the cross-verification of results. Both models benefit from a long validation record (e.g. Schepers, 2007; Schepers et al., 2018a, 2021a).

While the BEM method has to rely on engineering sub-models to account for dynamic inflow effects, AWSM's detailed wake modelling allows predicting the unsteady wake evolution with high fidelity (Boorsma et al., 2020). In AWSM the wake

is modelled by vortex filaments accounting for both trailed and shed vorticity, and the induced velocity at each point in space is estimated via the Biot-Savart law. Therefore, unlike blade-resolved CFD and experiments, the instantaneous induced velocities at the blade lifting lines are readily available without the uncertainty associated with induction extraction techniques (Schepers et al., 2018b). This makes the local induced velocities obtained with AWSM directly comparable to those of BEM facilitating the results verification. In this work, a special AWSM version that neglects the contribution of shed vortices in the induced

velocity calculations (both at the lifting lines and at the wake points) has also been developed aiming to gain insight into the physical effect of shed vortices on the results presented in Sect. 3.

## 2.2   Dynamic inflow models description

This section provides a brief description of the dynamic inflow models for BEM considered in this work. More detailed information on their derivation and parameter tuning can be found in the cited references, while their implementation in AM

is addressed in Sect. 2.3.

### 2.2.1   ECN

The ECN model was developed in the 90s by Snel and Schepers (1994) and consists of a semi-empirical correction to the classical momentum theory. In particular, a first-order term is added to the axial momentum equation to model the delay in the induction field response. Therefore the axial momentum equation in axisymmetric conditions writes:

$$2\rho A_{ann} \left[ f_a \frac{du}{dt} R + u \left( V_0 - u \right) \right] = N_b F_x \tag{1}$$

with $\rho$ being the air density, $A_{ann}$ the annulus area, $f_a$ a function of the radial position, $R$ the rotor radius, $u$ the annulus axial induced velocity, $t$ the time, $V_0$ the free-stream wind speed, $N_b$ the number of blades, and $F_x$ the local axial force exerted on the annulus.

The induction field delay is proportional to the function $f_a$, which is derived analytically under the assumption of a cylindri-

cal wake. This implies that wake expansion effects, which are important up to rated conditions, are neglected. Moreover, the radial evolution of the time delay plotted in fig. 2a has always been refuted by numerical and experimental evidence, and the presence of only one time constant does not allow to differentiate the fast decay due to near wake effects from the slower one induced by the far wake (Sørensen and Madsen, 2006).





### 2.2.2 Øye

The Øye dynamic inflow model was also developed in the 90s (Øye, 1990) but, contrarily to the ECN model, it uses two empirical first-order filters to correct the quasi-steady induction value obtained by solving the standard momentum equations. The corrected induced velocity is found by solving the following system for $u$:

$$y + \tau_1 \frac{dy}{dt} = u_{QS} + 0.6\,\tau_1\,\frac{du_{QS}}{dt} \tag{2}$$

$$y = u + \tau_2 \frac{du}{dt} \tag{3}$$

with $\tau_1$ being the slow time constant, $\tau_2$ the fast time constant, $t$ the time, $u_{QS}$ the axial induced velocity obtained by solving the steady-state momentum equations, and $u$ the corrected axial induced velocity.

The two time constants provide more accurate modelling of the unsteady induction response and they were calibrated with a simple vortex ring wake model of a uniformly loaded AD undergoing step changes of $C_T$. The slow constant ($\tau_1$), representative of far wake effects, only depends on the loading whereas $\tau_2$, which accounts for the near wake, has a quadratic dependency on 135 the radial location (the exact expressions can be found in Snel and Schepers, 1994).

A slight modification to the original formulation that improves the modelling of the induction response to coherent wind gusts was recently proposed by Berger et al. (2022). Their suggestion is to apply the slow filter in eq. 2 to the far wake velocity $u_{FW} = V_0 - 2u$ rather than $u$. Note that for a constant $V_0$ the modified formulation yields the same results as the classical one. Therefore, adapting the implementation would not affect the results of Sect. 3.1 and 3.2 and it is left for future works.

### 2.2.3 TUD-VR

The TUD-VR model was developed by Yu et al. (2019) and it was obtained by approximating the response of a uniformly loaded actuator disc to step changes of $C_T$ by means of two indicial functions. A correction model for the quasi-steady induced velocity was then derived from the theory of linear systems by expressing the Duhamel integral in differential form. Despite its different derivation, the quasi-steady induced velocity correction concept is quite similar to the Øye model, and it requires 145 solving the following system for $u$:

$$u = u_{QS} - \frac{1}{2}(c_1 + c_2) \tag{4}$$

$$\frac{dc_1}{dt} - \omega_1 \frac{V_0}{R} c_1 = 2\beta \frac{du_{QS}}{dt} \tag{5}$$

$$\frac{dc_2}{dt} - \omega_2 \frac{V_0}{R} c_2 = 2\,(1-\beta)\,\frac{du_{QS}}{dt} \tag{6}$$

with $u$ being the corrected axial induced velocity, $u_{QS}$ the axial induced velocity from steady-state momentum equations, $c_1$ 150 and $c_2$ correction functions obtained by solving eq. 5 and 6, $t$ the time, $V_0$ the free-stream wind speed, $R$ the rotor radius, and $\omega_1$, $\omega_2$, and $\beta$ polynomial functions of rotor loading and radial position. The coefficients for $\omega_1$, $\omega_2$, and $\beta$ were calibrated on step $C_T$ variations of both a linear and a non-linear vortex ring model of a uniformly loaded AD (expressions can be found in Yu et al., 2019). Only the non-linear version ("TUD-VR") has been used in this work.



The model builds on two main assumptions. One is that the relation between rotor thrust and induction can be assumed linear, which is a good approximation at low loading but it gets poorer as the induction increases. The other, which is shared with both Øye and DTU models, is that the unsteady induction response of a uniformly loaded AD represents well what happens in real rotors. Although the quality of this assumption heavily depends on the specific rotor design and operating conditions, tuning the parameters on a uniformly loaded disc makes these models applicable to any generic cases.

### 2.2.4   DTU

The DTU model for dynamic inflow was recently presented by Madsen et al. (2020) and implemented in HAWC2. Similar to the previous models, an empirical correction for the quasi-steady induced velocity was developed based on step loading changes of a uniform AD, this time modelled in CFD. The correction is carried out via two first-order filters leading to the following expression for $u$:

$$u = u_{QS} - \Delta u \left[ A_1 \exp \left( \frac{-t f_1}{\tau_1} \right) + A_2 \exp \left( \frac{-t f_2}{\tau_2} \right) \right] \tag{7}$$

with $u$ being the corrected axial induced velocity, $u_{QS}$ the axial induced velocity from steady-state momentum equations, $t$ the time, $A_1$ and $A_2$ two model constants, $f_1$ and $f_2$ linear functions of the local induction, and $\tau_1$ and $\tau_2$ the two time constants assumed to be quadratic functions of the radial position.

Note that the DTU model was conceived for the polar BEM formulation of HAWC2 (Madsen et al., 2020), and its adaptation to a conventional BEM solver featuring an annulus formulation (as in AM) might affect the results in non-uniform inflow conditions.

### 2.3   Dynamic inflow models implementation

In AM's BEM solver the BEM equations for each control element in each annulus are solved implicitly by means of the iterative Newton-Raphson algorithm. When the ECN dynamic inflow model is used a term depending on the first derivative of the annulus induction (an average between the local blade element inductions) is added to the axial momentum equation (eq. 1). This couples the BEM equations of the different blades making it hard to guarantee the local convergence at each element when non-uniform inflow conditions are considered (Boorsma et al., 2020).

Using the two-constant models listed in Sect. 2.2 avoids such a shortcoming. All these models, indeed, correct the quasi-steady axial induction value obtained by solving the steady-state momentum equations and they are applied outside the iterative solution loop. Thanks to their similarities a common implementation strategy has been devised following the process depicted in fig. 1.

Indicating the generic annulus with subscript $j$ and the current time instant with superscript $t$, the induced velocity correction ($\Delta u_j^t$) can be defined as:

$$\Delta u_j^t = \overline{u}_j^t - \overline{u}_{QS_j}^t \tag{8}$$





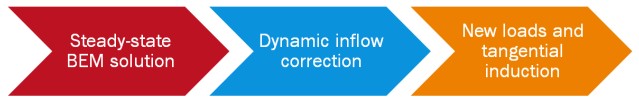

**Figure 1.** Process of induced velocity correction by the dynamic inflow model. Note that the "steady-state BEM solution" item only refers to the use classical momentum theory and airfoil unsteady effects are included.

with $\overline{u}_j^t$ being the annulus average axial induced velocity accounting for dynamic inflow effects, and $\overline{u}_{QS_j}^t$ the corresponding
value from steady-state momentum theory (i.e. accounting for skewed wake effects but not dynamic inflow).

Selecting the Øye, TUD-VR, or DTU dynamic inflow model only changes the way the induced velocity correction is computed. In fact, the routine that applies the induced velocity correction is common to all models and it just requires the annulus radius ($r_j$) and the annulus average free-stream axial wind velocity ($\overline{V}_{0_j}^t$) to be provided as inputs, along with $\overline{u}_{QS_j}^t$. The correction expressions for each model are reported in Appendix A.

Note that in non-uniform conditions the value of $\overline{V}_{0_j}^t$ depends on the reference velocity chosen for each blade. The local wind speed at each blade control point is used in AM and the average of these values yields $\overline{V}_{0_j}^t$ (Boorsma et al., 2020). To include the loading dependency of the model parameters, the quasi-steady annulus average axial induction factor is evaluated ($\overline{a}_{QS_j}^t = -\frac{\overline{u}_{QS_j}^t}{\overline{V}_{0_j}^t}$) before computing the correction according to the selected model.

Following AM's BEM solver philosophy, an annulus approach has been chosen to compute and apply the dynamic inflow
correction. Therefore all the steps highlighted in this section are repeated for each control annulus assuming complete radial independence. This is just one of the various implementation choices possible (Schepers et al., 2018a) and it might not be the optimal one (provided that a general optimum exists). These choices have no impact on axial uniform inflow cases (Sect. 3.1 and 3.2), but they do affect generic non-uniform cases. Nevertheless, the consistency of all the implementation choices for the different models guarantees the validity of this comparative study where the relative performance is assessed.

Once $\Delta u_j^t$ is known, the new annulus average axial induced velocity is obtained from eq. 8. Finally, in order to evaluate the new tangential induction (resulting from the coupling of the momentum equations) and update the loads on each element, it is assumed that the same correction that applies to the annulus average value (i.e. the average among the blades) also applies to the annulus axial induced velocity of each individual blade.

## 2.4 Simulations setup description

Within the STRETCH project, a 220m diameter variable speed rotor design rated at 12MW has been developed following the current design process for large rotor blades to provide the research community with a realistic reference turbine. The rotor has been designed for a rated wind speed of $10.5m/s$ with an average induction close to $1/3$ at the optimal tip speed ratio. For this rotor, a new variable speed and individual pitch controller has been designed by TNO. The resulting state-of-the-art offshore wind turbine design has been used to assess the impact of the aerodynamic modelling activities carried out within the project.

The dynamic inflow model verification and benchmarking campaign has been performed in two steps. At first, standard dynamic inflow tests based on prescribed blade pitch actuations in axial uniform wind conditions have been considered, using



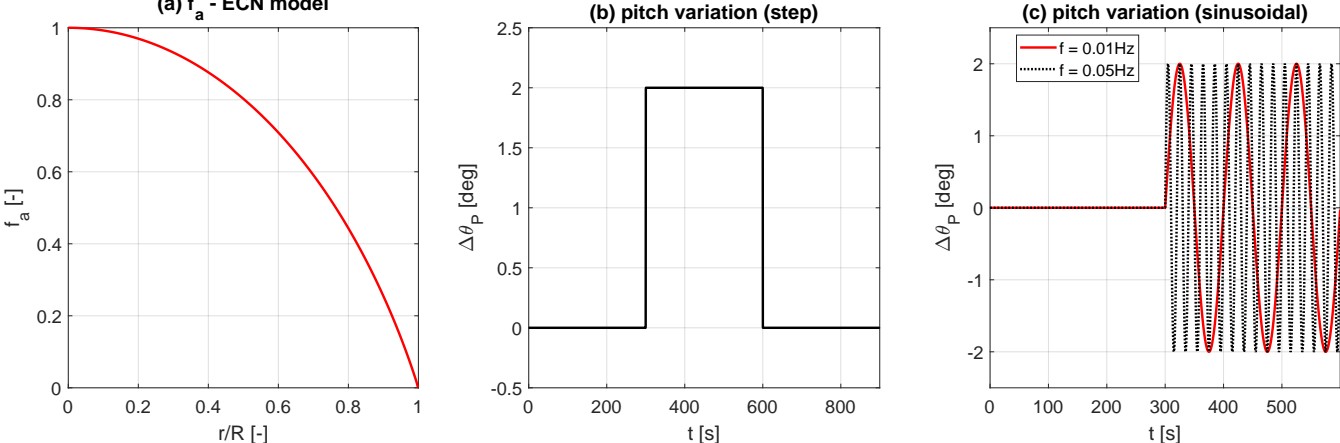

**Figure 2.** Radial evolution of the ECN model parameter $f_a$ (a); blade pitch time history for the pitch step case (b); blade pitch time history for the two sinusoidal pitch cases (c).

a rigid turbine model (standalone AM simulations) and neglecting the shaft tilt angle to avoid skewed wake effects (Sect. 3.1 and 3.2). These simple cases allowed isolating the effect of the dynamic inflow models on BEM results verifying their predictions by comparing them with AWSM. Secondly, a standard fatigue load case (DLC 1.2) has been run with a complete
aeroelastic model of the STRETCH rotor, trying to assess the practical impact of BEM dynamic inflow models on design driving simulations (Sect. 3.3). The numerical setup for the two steps is described in the following subsections.

### 2.4.1 Simple conditions

As a first step, standalone AM simulations featuring a rigid model of the STRETCH rotor in axial uniform wind conditions have been performed. The rotor speed has been kept constant while different collective blade pitch time histories have been
prescribed to trigger dynamic wake effects. The shaft tilt angle and the tower influence have been neglected to simplify the results interpretation. On such long blades (107m) the operating torsion deformation has a significant impact on the aerodynamic performance as it directly affects the angle of attack and thus the rotor loading. One way to account for this effect with a rigid blade model is to assess the spanwise distribution of the average torsion deformation at the desired operating conditions with an aeroelastic tool, and use that information to modify the prescribed aerodynamic twist distribution accordingly. This
approach has been followed in all the rigid simulations performed.

Both BEM and AWSM simulations have been performed using the same time step, which corresponded to a little less than $10^o$ of azimuthal blade rotation. A total wake length of six diameters has been modelled in AWSM so as to make sure that the far wake effects affecting the slow decay of the induction response could be correctly captured. The wake convection has been modelled as free for the first two diameters downstream of the rotor plane, while each blade vortex average induction
at the free-to-fixed interface has been used for the far wake convection. AWSM simulations have been run using 21 control elements along the span following a cosine distribution. For the dynamic stall, the first-order model from Snel (1997) has been





used in compliance with the guidelines provided in Boorsma and Caboni (2020). As mentioned in Sect. 2.1, some simulations have also been repeated using a special version of AWSM that does not include the shed vorticity contribution in the induced velocity calculations in order to highlight the physical effect of shed vorticity by comparison with the standard AWSM results.

For the BEM simulations, the number of control elements has been set to 30, all equally spaced except at the tip where the spacing between the last two elements is halved. Prandtl's correction has been applied at the blade root and tip, and the Beddoes-Leishman model (Hansen et al., 2004) has been used for the dynamic stall (unless diversely specified) following the guidelines given in Boorsma and Caboni (2020). All the settings have been kept constant while switching between the different dynamic inflow models.

### 2.4.2   DLC 1.2

In order to assess the practical impact of dynamic inflow models for BEM on wind turbine design load simulations, the fatigue DLC 1.2 from the International Electrotechnical Commission (IEC) 61400-1 standard  has been selected. To be as representative as possible of real design cases the same turbine aeroelastic model used for the STRETCH rotor design calculations has been used, without the simplifications made for the rigid simulations (Sect. 2.4.1). The load case calculations have been

performed with PhatAero-BEM (Boorsma et al., 2020), including the flexibility of blades and tower, and switching between the different dynamic inflow models while keeping all other inputs the same. Besides the dynamic inflow models, the main BEM sub-models that have been used in these simulations are: the ECN model for yaw (Schepers, 1999); Prandtl's root and tip corrections; Snel's dynamic stall model (Snel, 1997); a potential flow model for the tower influence.

The blade span has been discretized using 28 control points for the aerodynamic calculations. The structural solver settings

followed the standard guidelines of Phatas (Lindenburg and Schepers, 2000) and a time step of 0.025s, common to both the aerodynamic and the structural solver, has been used in all runs. Intervals of 640s have been simulated, always skipping the first 40s to exclude the initial settling of the turbine operational response.

The turbulent inflow for DLC 1.2 has been generated following the IEC Normal Turbulence Model specifications (class IB) using the SWIFT wind field generator (Winkelaar, 1992) based on Kaimal's spectrum. The full load case including six seeds

per wind speed was originally run using the original STRETCH turbine controller. However, using different dynamic inflow models with an active controller results in different regulations of blade pitch and rpm for the same turbulent wind field making the results hardly comparable and the observed differences very difficult to interpret. The outcome of this activity is briefly commented in Appendix B.

With the aim of benchmarking the dynamic inflow models for BEM, it has been decided to give up some of the complexity

for the sake of interpretability. Therefore, the rotor speed and blade pitch time histories have been prescribed while keeping all other simulation settings unchanged. The following approach has been pursued:

1. The full DLC 1.2 has been simulated with a steady BEM solver without dynamic inflow models (hereinafter named MT after Momentum Theory), and using a collective pitch version of the standard STRETCH rotor controller. The load case has been run considering 6 seeds for each wind speed in the power curve following the IEC standard requirements.





2. To reduce the number of simulations, a subset of wind speeds where dynamic inflow effects are of interest has been selected from the full dataset of point 1.

    3. For each selected wind speed, the seed closest to the average (among the six) in terms of 1Hz-equivalent out-of-plane blade root bending moment has been selected.

    4. For the selected wind speeds and seeds, reference rotor speed and blade pitch time histories have been extracted from
the MT simulations of point 1.

    5. The selected seeds at the selected wind speeds have been re-simulated with the different dynamic inflow models prescribing the reference rotor speed and blade pitch time histories of point 4.

The time histories of the rotor speed and blade pitch have been prescribed by exploiting a special controller developed within the AVATAR project (Schepers et al., 2018c), which adjusts the blade pitch angle and regulates the torque to follow the target
blade pitch and rotor speed values specified by the user. In its current version, such a controller does not support individual pitch control and this is why a collective pitch version of the STRETCH rotor controller has been adopted in point 1.

## 3   Results

The main results of the dynamic inflow verification campaign are presented in this section, starting from the collective pitch steps around rated wind conditions (Sect. 3.1) and then passing to the sinusoidal pitch variations both near and above rated
conditions (Sect. 3.2). The aeroelastic simulations results for DLC 1.2 are discussed in Sect. 3.3.

### 3.1   Near-rated pitch steps

Blade pitch steps represent the traditional test case to investigate the performance of dynamic inflow models (e.g. Snel and Schepers, 1994; Sørensen and Madsen, 2006; Berger et al., 2020). This is because the slow response of the induction to sudden pitch changes leads to load overshoots that may affect the fatigue life of components. These pitch variations are often designed
to be representative, at least to some extent, of fast controller actions.

    A pitch step case has been considered in this work as well. The test selection has been inspired by a numerical case simulated in the international code comparison round of the IEA Task 29 (Schepers et al., 2018b), where a very fast pith variation was imposed to the AVATAR turbine. The very same pitch step variation has been prescribed in this case, starting from the design blade pitch value, suddenly raising it by $2^o$ towards feather, holding it for 300s, and finally returning to the initial condition
(fig. 2b) with a maximum pitch rate of $\pm 10^o/s$. The fast pitching rate helps highlight differences between the models.

    The test has been run near rated conditions (10 m/s wind speed) where dynamic inflow effects are expected to be most significant. Indeed, pitch control actions only start occurring once the rated power is reached, but the higher the wind speed the lower the induction factor leading to weaker wake unsteadiness. Therefore, the only standard operating region where pitch actions are likely to occur at high rotor induction (typically close to the Betz optimum) is the one near rated power. Rotational

**Figure 3.** Normalized axial induction factor response to the pitch steps near rated conditions at four different radial locations: (a) 40% span; (b) 60% span; (c) 80% span; (d) 95% span.

speed variations below rated conditions might also play a role, but they are usually much slower due to the great inertia of large variable-speed rotors and thus trigger little unsteadiness.

In order to isolate the unsteady behaviour from discrepancies in the steady-state values (always present between BEM and AWSM results, albeit small), the time histories of all quantities have been normalized as follows:

$$X^*(t) = \frac{X(t) - X_{0^o}}{X_{0^o} - X_{2^o}} \tag{9}$$

with $X(t)$ being the time history of a generic quantity, $X_{0^o}$ and $X_{2^o}$ the steady-state values before and after the feathering step, and $X^*(t)$ the resulting normalized time history. Despite the normalization helping to better compare the results of different models, the radial dependency of the induction time histories hinders a synthetic visualization.





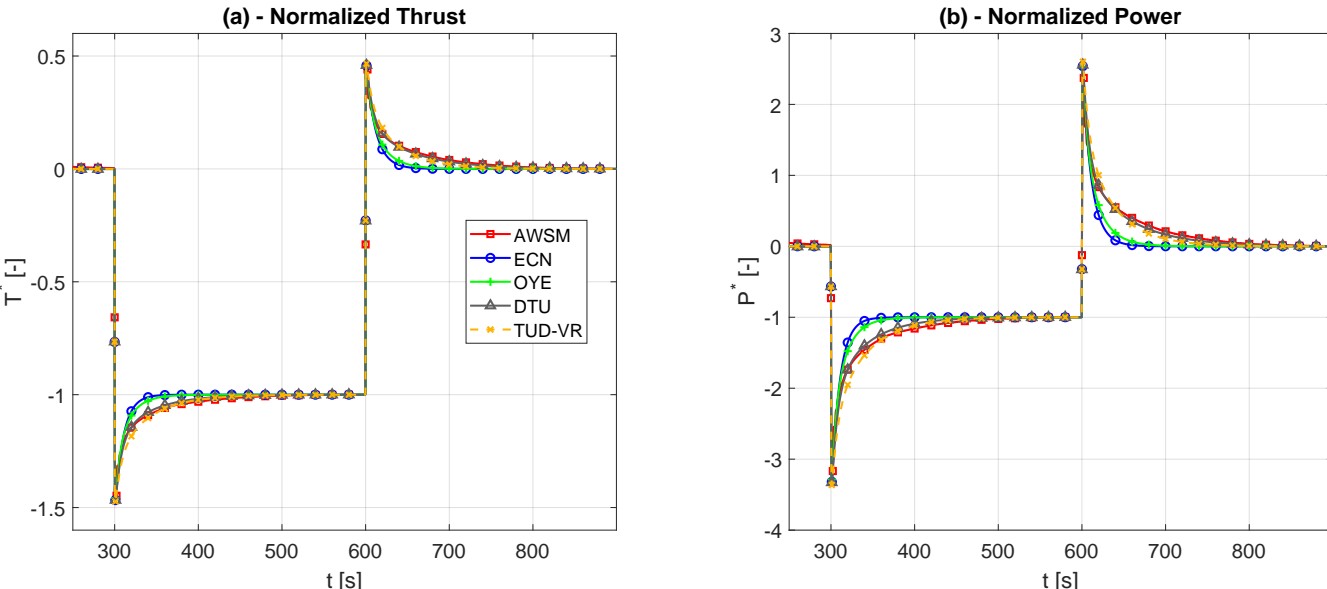

**Figure 4.** Normalized rotor aerodynamic thrust (a) and power (b) response to the blade pitch steps near rated conditions.

The axial induction responses, normalized via eq. 9, at four spanwise locations are shown in fig. 3. As expected, the ECN model predictions provide the worst match with AWSM, with the single time constant leading to a faster induction reconfigu-
ration in both pitch step directions that confirms the findings of previous studies. This is especially true in the tip region where $f_a$ approaches zero (fig. 2a) and the ECN model response is almost quasi-steady. Slightly better predictions have been obtained with the Øye model, which gives similar results to the ECN model in the inboard part of the blade but finds a slower decay close to the tip. The TUD-VR and DTU models provide the best match with AWSM for the case considered. Their predictions appear almost equivalent and both very similar to the free vortex results until the blade tip, where a slightly faster decay than
AWSM is observed. It is important to notice that, due to the very fast pitch rate, the AWSM results show a sudden spike in the induction response immediately after the step (i.e. around 300s and 600s) that cannot be replicated by the BEM models. Section 3.1.1 shows this behaviour to be a result of the sudden change of the blade bound circulation induced by the pitch maneuver, which produces strong shed vorticity in the near wake.

The normalized aerodynamic rotor thrust and power are plotted in fig. 4. Their behaviour is consistent with the observations
made on the induction field, with Øye and ECN models predicting a faster decay after the overshoot than TUD-VR, DTU, and AWSM. Although difficult to spot in the figures, all BEM models slightly overestimate the overshoot peak, especially in terms of power (where peaks are larger). This is a direct consequence of the induction spike caused by the shed vorticity that sharply increases the instantaneous induction right after the step and hence reduces and delays the overshoot.



### 3.1.1 The shed vorticity effect

As discussed in the previous section, a sudden peak in the induction response right after the pitch step and before the start of the exponential build-up has been observed. Zooming around the time interval of the loading step reveals that thrust and power responses exhibit a staircase behaviour (fig. 5c and d) similar to the one observed in the IEA Task29 simulating pitch steps on the AVATAR turbine (Schepers et al., 2018b). To build on what was observed back then, a special AWSM version disregarding the shed vorticity contribution in all induced velocity calculations has been developed. Although neglecting the presence of

shed vortices violates Helmoltz's second theorem on vorticity, this special AWSM version is only used in comparison with the standard one allowing to gain insight into the impact of the shed vortices in the wake.

The plots in fig. 5 also include the TUD-VR results obtained with two different dynamic stall models: the Beddoes-Leishman model (Hansen et al., 2004), which takes the Theodorsen effect on airfoils into account; and the Snel first-order model (Snel, 1997) that does not. This allows assessing the impact on induction results of modelling the shed vorticity at an airfoil level.

Such a comparison is synthesized in fig. 5, where the normalized axial and tangential induction factors at $60\%$ of the span are shown as an example (fig. 5a and b), along with the normalized rotor thrust and power (fig. 5c and d).

The axial induction plot clearly shows that without shed vorticity the peak disappears and the staircase effect becomes less pronounced. This is even more evident in the tangential induction plot. In agreement with Task29 findings, the lack of shed vorticity translates into a greater and earlier overshoot. Concerning the BEM results with the two dynamic stall models,

it appears that the shed vorticity contribution in the Beddoes-Leishman model moves the predictions in the right direction, partially reducing and delaying the load overshoot as expected. However, this effect is much smaller than what is observed in AWSM partly because the Beddoes-Leishman correction, in its current implementation, is only applied to the airfoil coefficients and only indirectly affects the induced velocities. Applying a shed vorticity correction to the induced velocities may improve the results, but such an investigation is left for future research. Another aspect contributing to the observed discrepancy is that

the vortex shedding is modelled in a 2D fashion in BEM (using the Beddoes-Leishman model) assuming radial independence for each annulus. In other words, each airfoil is only affected by its own shed vorticity and it is blind to what happens in the neighbouring elements. Whenever highly unsteady conditions are concerned, this approximation is expected to make the overall shed vorticity effect in BEM less significant than in reality since only the airfoil scale is modelled.

One more interesting remark can be made by observing the tangential induction plot (fig. 5b). The BEM models predict a

small overshoot right after the pitch step, which is purely an effect of the coupling between axial and tangential momentum equations, i.e. correcting the quasi-steady axial induction leads to a new tangential induction value that guarantees a new equilibrium (Sect. 2.3). A tangential induction peak was also observed experimentally by Berger et al. (2021), and a similar trend can be recognized in the AWSM results as well once the initial oscillations induced by the shed vorticity are damped out. Although not reported here, the simulation of a slower pitch step where shed vorticity effects were negligible has produced

similar tangential induction results for both BEM and AWSM, confirming the presence of a peak that compensates for the lower axial induction immediately after the pitch action.





**Figure 5.** Zoom in the time interval near the loading step to highlight the effect of shed vorticity in the time history of: (a) the normalized axial induction at 60% span; (b) the normalized tangential induction at 60% span; (c) the normalized aerodynamic thrust; (d) the normalized aerodynamic power.

### 3.2 Sinusoidal pitch variations

One of the main drawbacks of considering pitch steps to verify the performance of different dynamic inflow models is the difficulty of visualizing results in a compact yet comprehensive way. Prescribing sinusoidal collective blade pitch variations
helps a lot in that sense. Observing that the spectrum of the induction response to a mono-harmonic blade pitch variation is dominated by the component at the pitch actuation frequency, the resulting unsteady axial induction can be fully characterized by the amplitude and phase of that harmonic. This allows characterizing all radial locations in one pair of plots.

Sinusoidal variations were also considered by Yu et al. (2019), but there the use of a uniformly loaded actuator disk allowed a direct prescription of the thrust coefficient variations. Here more detailed lifting line rotor models are considered instead,





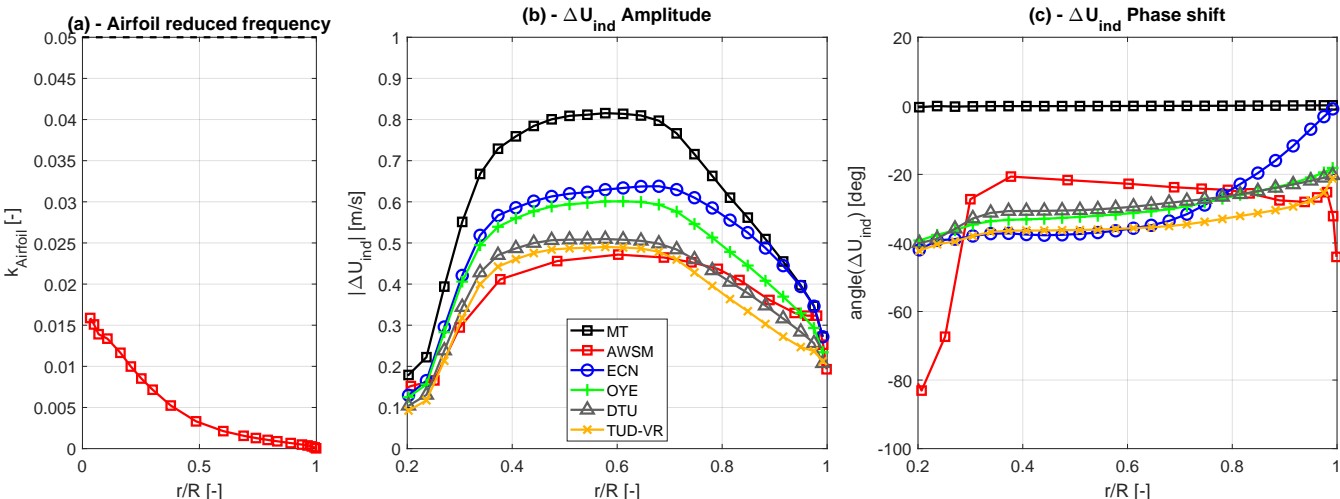

**Figure 6.** Spanwise distributions for the $10mHz$ sinusoidal pitch variation near rated conditions: (a) of the airfoil reduced frequency; (b) of the axial induced velocity variation amplitude at the blade pitch frequency; (c) of the axial induced velocity variation phase at the blade pitch frequency, with respect to the phase of the prescribed pitch time series.

therefore the rotor loading is only varied indirectly through the blade pitch angle. Due to the non-linear dependency of $C_T$ from the blade pitch angle (especially at high loading), and the non-uniform spanwise loading of a real rotor, the present results are hardly comparable to those of Yu et al. (2019) though some conclusions are alike.

Both near- and above-rated conditions have been considered in this case, with wind speeds of 10m/s and 15m/s and average angles of attack of $\sim 7^o$ and $\sim 4^o$, respectively. A sinusoidal blade pitch variation with respect to the design pitch setting has

been prescribed, always with an amplitude of $A_p = 2^o$ that triggers significant wake unsteadiness while guaranteeing to avoid stall effects in the last $75\%$ of the blade span. Two different frequencies have been considered: $f_p = 10mHz$; and $f_p = 50mHz$. These correspond to a rotor reduced frequency (defined as: $k = \frac{f_p R}{V_0}$; with $f_p$ the pitch frequency, $R$ the rotor radius, and $V_0$ the free-stream wind speed) of $k \simeq 0.11$ and $k \simeq 0.55$ near rated, and $k \simeq 0.07$ and $k \simeq 0.36$ above rated conditions. The frequencies have been selected multiple of the simulation time step to prevent leakage effects in the spectra that have been

obtained via Fast Fourier Transform.

The near-rated results for the two pitch frequencies are shown in fig. 6 and 7, respectively. The first plot in the two figures (6a and 7a) shows the spanwise evolution of the airfoil reduced frequency (defined as: $k_{airfoil} = \frac{\pi f_p c}{W}$; with $f_p$ the pitch frequency, $c$ the airfoil chord, and $W$ the airfoil effective wind speed). The mean effective wind speed found by AWSM during a pitch cycle has been used to compute $k_{airfoil}$. Even in the higher frequency case, the values remain below the conventional

$5\%$ threshold from $25\%$ span onwards, with maximum values at the root never exceeding $10\%$. This guarantees that airfoil unsteady effects are limited and standard polars can be relied upon in all the cases considered.

Looking at the amplitude versus span plots for the two frequencies (fig. 6b and 7b) it can be noticed how dynamic inflow effects reduce the axial induced velocity oscillations with respect to the quasi-steady momentum theory prediction (the black





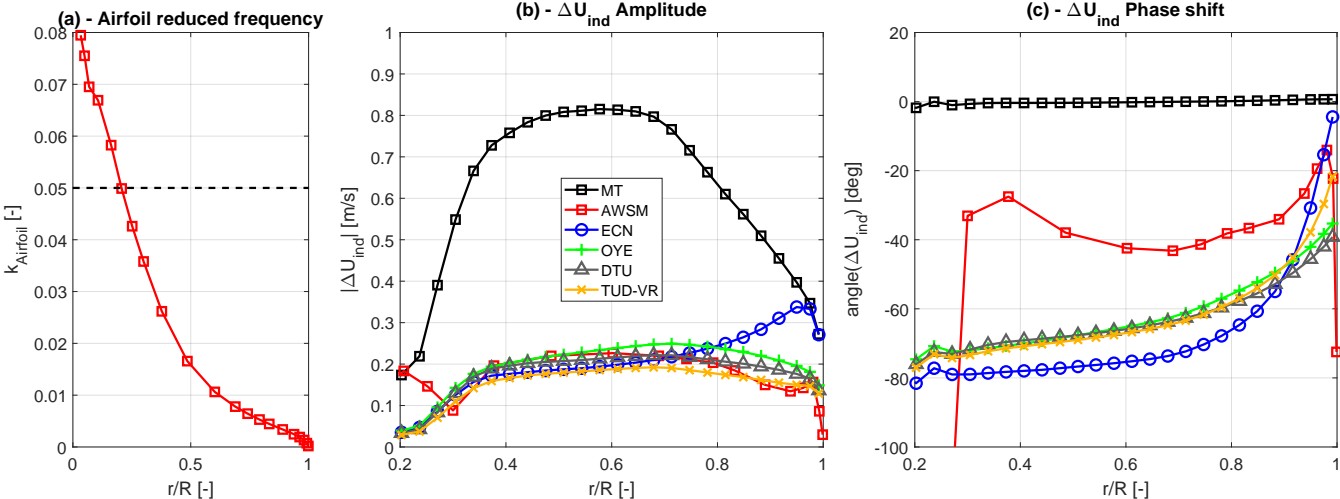

**Figure 7.** Spanwise distributions for the $50\,mHz$ sinusoidal pitch variation near rated conditions: (a) of the airfoil reduced frequency; (b) of the axial induced velocity variation amplitude at the blade pitch frequency; (c) of the axial induced velocity variation phase at the blade pitch frequency, with respect to the phase of the prescribed pitch time series.

line in the figures), especially for the higher pitch frequency. Compared to AWSM, ECN and Øye models tend to slightly

overestimate the oscillation amplitudes at the lower frequency (fig. 6b). A better agreement is found for the faster pitch variation (fig. 7b), except for the ECN model prediction in the outboard sections that rises to reach the quasi-steady value at the tip. As in the pitch step case, the TUD-VR and DTU models provide very similar results that match the AWSM predictions well.

Figures 6c and 7c show the radial evolution of the phase shift between the axial induced velocity oscillation (at the pitching frequency) and the blade pitch actuation signal. Greater differences can be observed in these plots, with an average phase

difference with respect to AWSM of about $30^o$ throughout the span for the fast frequency case. Smaller discrepancies are also noticeable for the slower pitch variation. Besides the ECN line that always goes to zero at the tip (i.e. quasi-steady behaviour), the discrepancies between AWSM and the other BEM models are not fully understood. Searching for the critical parameters affecting such a phase shift, it was found that a reduction of the far wake correction in the TUD-VR and DTU models generally improves the phase match with AWSM, although with conflicting effects on the amplitude (Schepers et al., 2021b). Shed

vorticity also contributes to the observed phase differences tending to reduce the phase delay in AWSM (as discussed in Sect. 3.2.1) and thus increasing the mismatch with BEM results.

Very similar conclusions can be drawn from the above-rated simulations. The slower blade pitch results have not been reported here, as dynamic inflow effects were barely visible due to the low rotor loading. The usual plots for the higher frequency case are shown in fig. 8. As expected, the quasi-steady induction variation amplitudes are smaller than the corresponding near-

rated case, leading to milder dynamic wake effects (fig. 8b). In terms of models performance, the same comments made for the rated case apply to these results as well, with all two-constant models behaving similarly and overestimating the phase shift compared to AWSM.



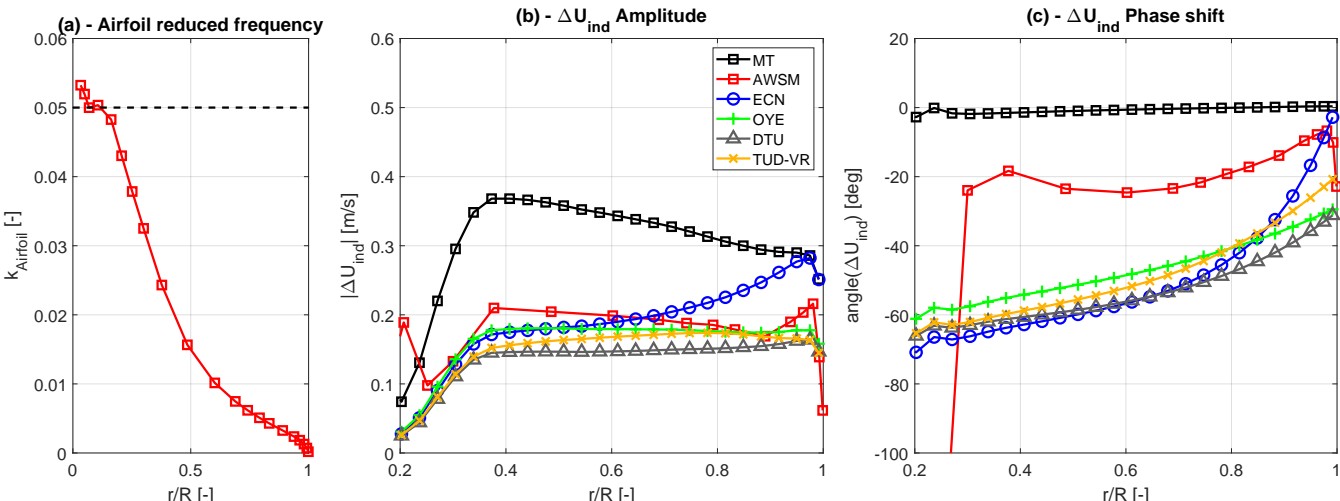

**Figure 8.** Spanwise distributions for the $50mHz$ sinusoidal pitch variation above rated conditions: (a) of the airfoil reduced frequency; (b) of the axial induced velocity variation amplitude at the blade pitch frequency; (c) of the axial induced velocity variation phase at the blade pitch frequency, with respect to the phase of the prescribed pitch time series.

### 3.2.1 Shed vorticity effect

Similar to the pitch step case, additional simulations have been carried out to investigate the effects of shed vorticity on the
induction response to sinusoidal pitch variations in order to gain insight into the physics and better understand the observed differences between BEM and AWSM. Only the higher frequency case was considered since the intensity of shed vortices grows with the pitch rate. Moreover, results are shown for the above-rated conditions, where controller-induced blade pitch corrections are more frequent and significant.

The corresponding amplitude and phase plots are shown in fig. 9. As for the pitch step, the BEM lines (with the TUD-VR
model for dynamic inflow) accounting or not for shed vorticity in the dynamic stall model have been added, along with those of AWSM with and without shed vorticity. Figure 9a shows how shed vortices tend to slightly reduce the induced velocity variation amplitude throughout the span. Whereas in terms of phase, the no-shed vorticity line shows a larger (i.e. more negative) phase shift coming closer to the BEM predictions.

Switching between Snel and Beddoes-Leishman dynamic stall models to include the Theodorsen effect on the airfoil coef-
ficients barely affects the BEM results, with minor differences noticeable in the inboard part of the blade only. Despite their very small extent, these discrepancies are a bit puzzling because unlike the pitch step case of Sect. 3.1.1, where the use of the Beddoes-Leishman model improved the match with the standard AWSM, here contrasting effects are observed for the amplitude and the phase. Furthermore, using the Snel model appears to have an opposite effect on the BEM predictions than neglecting the shed vorticity contribution in AWSM. A possible reason contributing to this inconsistency might be an unwanted

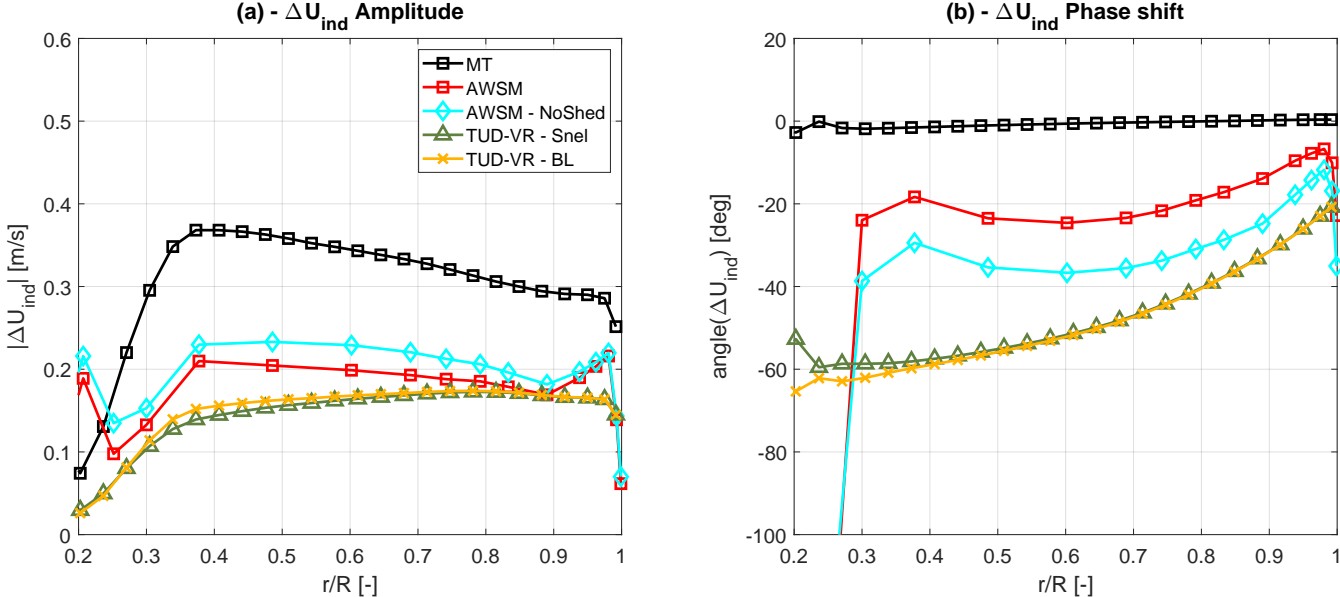

**Figure 9.** Effect of the shed vorticity on the $50 mHz$ sinusoidal pitch variation above rated conditions: (a) on the distribution of the axial induced velocity variation amplitude at the blade pitch frequency; (b) on the distribution of the axial induced velocity variation phase at the blade pitch frequency, with respect to the phase of the prescribed pitch time series.

effect of the Snel model that was found by Boorsma and Caboni (2020). When the airfoil lift curve deviates from the theoretical potential flow slope, the model slightly acts in attached flow regions where it should not.

### 3.3  Impact of dynamic inflow models on the BEM results for DLC 1.2

To assess the impact of the different dynamic inflow models for BEM on a practical design load case, the procedure outlined in Sect. 2.4.2 has been followed. Being DLC 1.2 a fatigue load case, the main load time histories have been transformed into 1Hz equivalent load cycles following the guidelines of the IEC 61400-1. Figure 10 shows the comparison of the equivalent out-of-plane bending, in-plane bending, and torsion moments at the root of blade 1, along with the tower base fore-aft moment for several wind speeds along the turbine operating curve. The values have been normalized by the corresponding equivalent moments obtained by running steady-state BEM simulations (MT), i.e. without dynamic inflow models. This helps highlight the relative differences in various operating conditions.

In general, the use of a dynamic inflow model leads to higher equivalent loads up to $\sim 10\%$ for the out-of-plane blade root bending moment (fig. 10a). The tower base moment (fig. 10d), which is driven by the rotor thrust and thus includes the contributions of all the blades, shows up to $\sim 20\%$ greater loads. Lower increments are observed in the equivalent torsion moment at the root of blade 1 as well (fig. 10c). Instead, the in-plane root bending moment (fig. 10b) is almost insensitive to the dynamic inflow models as it is largely dominated by the gravitational load cycles.




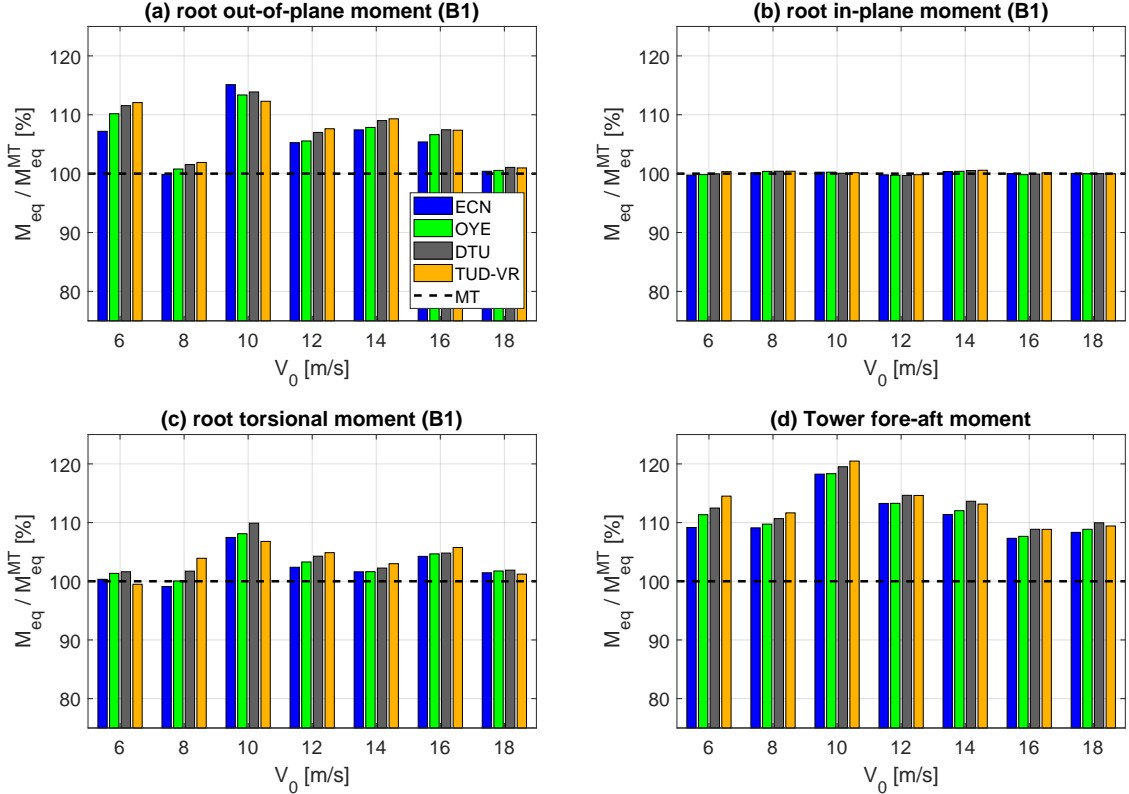

**Figure 10.** Comparison of 1Hz equivalent moments obtained with the different dynamic inflow models for different wind speeds in DLC 1.2. The equivalent out-of-plane (a), in-plane (b), and torsional (c) moments at the root of blade #1 are reported along with the fore-aft moment at the tower base (d). All equivalent moments are normalized by the value obtained without using dynamic inflow models (MT).

Provided that similar wind speed, rotor speed, and blade pitch time histories have been prescribed, the general equivalent load increase can be attributed to the damping of the induced velocity fluctuations that results in greater aerodynamic load variations when dynamic inflow models are considered. The maximum equivalent load increase is always located around rated conditions (at 10m/s in this case) because there is where the blade pitch regulation starts and the rotor loading is still high. As the wind speed increases above rated conditions the average induction lowers leading to milder dynamic wake effects despite the relevant blade pitch regulation. The opposite is true below rated conditions, where the high tip speed ratio makes the rotor more sensitive to dynamic wake effects but the constant blade pitch reduces the chances of sharp loading variations occurring. In fact, unless considerable wind gusts occur, the large inertia of a more than 200m diameter rotor makes rotational speed variations less prone to trigger strong dynamic inflow effects.

Provided that a dynamic inflow model is used in the BEM simulations, the choice of the specific model seems to matter significantly less in terms of equivalent load predictions. The differences found among the models fall always within 5%, with





larger discrepancies found at the wind speeds where the induction is higher. The TUD-VR and DTU models yield similar predictions as in the uniflow inflow cases (Sect. 3.1 and 3.2), with the Øye model typically falling between those and the ECN model results.

Once more, these results have been obtained by prescribing the time histories of the rotor operating conditions to aid their
interpretation. The actual differences for a fully realistic load case including the original turbine controller and considering six seeds per wind speed have been found to be higher, especially below rated (as shown in fig. B1 in Appendix B). This is because different dynamic inflow models lead to different controller actions and thus different operating conditions.

## 4  Conclusions

As part of the aerodynamic modelling activities in the TKI STRETCH project, three new dynamic inflow models have been
implemented in AM's BEM solver: the Øye, the DTU, and the TUD-VR model. With these two-time-constant models, alternatives to the ECN model originally in AM, all the main dynamic inflow models available in the literature have been implemented. Sharing the same BEM solver and a common implementation strategy (Sect. 2.3) has reduced the uncertainty in the comparison of the different model predictions.

A 220m diameter state-of-the-art offshore wind turbine designed in the STRETCH project has been considered for this
comparative study. To facilitate the results verification against free vortex wake model predictions, a rigid rotor model in axial uniform inflow conditions has been considered first, prescribing conventional blade pitch steps as well as sinusoidal pitch variations. The latter offered a convenient way to visualize and compare the results. The effect of the blade shed vorticity on these uniform inflow cases has been investigated by exploiting a special version of AWSM, which neglects the contribution of shed vortices in the induced velocity calculations.

Analysing fast pitch steps near rated conditions (Sect. 3.1) has shown that two-constant dynamic inflow models reproduce the induction decay better than the ECN model, especially in the blade tip region, confirming the conclusions of previous studies (e.g., Schepers, 2007; Sørensen and Madsen, 2006; Pirrung and Madsen, 2018). In this study, the best agreement with the free vortex wake results for the whole blade span has been reached with the most recent TUD-VR and DTU models that have been found to yield very similar predictions. The Øye model results typically fell between those and the ECN model
predictions overestimating the speed of the induction recovery.

The strong vorticity shed as a result of the fast pitch steps has been found to cause a sudden induction peak that reduces and delays the load overshoots/undershoots in agreement with the findings of Schepers et al. (2014). A slight staircase effect due to the trailed vorticity has also been observed. Modelling shed vorticity effects on the airfoil coefficients with the Beddoes-Leishman model has been shown to improve BEM results, but to a marginal extent compared to what the AWSM predictions
with and without shed vorticity have highlighted. The 2D nature of the BL correction combined with the fact that it is only applied to the airfoil coefficients and not on induced velocities are indicated as the most likely causes for this. As a result of the coupling between the axial and tangential BEM equations, a tangential induction peak has been noticed as well confirming recent experimental evidence from Berger et al. (2021).



The sinusoidal blade pitch variation cases have confirmed the findings of the pitch steps, highlighting the ECN model's tendency to a quasi-steady behaviour near the blade tip. In terms of the induced velocity variation amplitudes, two-constant models (especially TUD-VR and DTU) have provided quite accurate predictions for all the cases considered. For the phase, large discrepancies between BEM and AWSM results have been found in the fast actuation cases instead. These are partly caused by the shed vorticity, although, unlike the pitch step case, the use of the BL model has not given any benefit to the BEM predictions leaving this point open to further investigations.

For the first time, the impact of dynamic inflow models on a fatigue design load case has been assessed quantitatively. Prescribing the same operating condition time histories to guarantee a fair comparison, a systematic increase of the equivalent loads with respect to a steady-state BEM formulation has been found with all dynamic inflow models. The hypothesis of dynamic inflow effects being most relevant around rated wind conditions, where blade pitching occurs at high induction, has been confirmed showing a 10-20% increase in 1Hz equivalent out-of-plane bending moment at both blade root and tower base. Differences among the dynamic inflow models have been found larger at high rotor loading, but always within 5%. A more realistic comparison including the turbine controller has revealed larger discrepancies between the models, but the differences in operating conditions hinder the interpretation of the results (Appendix B).

Overall, this work has confirmed the superiority of two-constant models compared to the ECN model, both for the better modelling of the induction decay and the easier implementation in an implicit BEM, which avoids local convergence issues in non-uniform inflow conditions. The targets mentioned in Sect. 1 have been reached. However, concerning the ultimate aerodynamic modelling goal of improving the accuracy of BEM design load calculations, this study suggests that further dynamic inflow model improvements, albeit possible, are expected to have limited impact compared to improvements in the treatment of non-uniform inflow conditions or modelling of shed vorticity effects. Therefore, these aspects shall be given priority in future works. The next step for this dynamic inflow modelling activity will be the implementation of the recently proposed Øye model modification that improves the modelling of wind gust-driven dynamic wake effects (Berger et al., 2022). Simulating DLC 1.2 with this new model may give a better idea of the relevance of turbulent wind variations on dynamic inflow effects on large rotors.

## Appendix A: new dynamic inflow models implementation

This appendix reports the expressions for the induced velocity correction used by the different two-constant dynamic inflow models that have been implemented in AM as described in Sect. 2.3. The expressions for each dynamic inflow model are presented in the following sections.

## A1  Øye model correction

The Øye model correction is computed with the following steps:

1. Evaluate the time constants $\tau_{1_j}^t(\overline{a}_{QS_j}^t, \overline{V}_{0_j}^t)$ and $\tau_{2_j}^t(\tau_{1_j}^t, r_j)$ according to the expressions reported in Snel and Schepers (1994).





2. Evaluate $y_j^t$ solving eq. 2 (with backward discretization).

3. Calculate the Øye model correction (from eq. 3):

$$\Delta u_j^t = \frac{y_j^t + \frac{\tau_{2_j}^t}{\Delta t}\overline{u}_j^{t-1}}{1 + \frac{\tau_{2_j}^t}{\Delta t}} - \overline{u}_{QS_j}^t \qquad (A1)$$

**A2 TUD-VR model correction**

The TUD-VR model correction is computed with the following steps:

1. Evaluate the local thrust coefficient $C_{T_j}^t(\overline{a}_{QS_j}^t)$.

2. Evaluate the model parameters $\beta_j^t(C_{T_j}^t, r_j)$, $\omega_{1_j}^t(C_{T_j}^t, r_j)$, and $\omega_{2_j}^t(C_{T_j}^t, r_j)$ from the polynomial expressions reported in Yu et al. (2019).

3. Evaluate $c_{1_j}^t$ and $c_{2_j}^t$ solving eq. 5 and 6 (with backward discretization).

4. Calculate the TUD-VR model correction (from eq. 4):

$$\Delta u_j^t = -\frac{1}{2}\left(c_{1_j}^t + c_{2_j}^t\right) \qquad (A2)$$

**A3 DTU model correction**

The DTU model correction is computed with the following steps:

1. Compute the model parameters $f_{1_j}^t(\overline{a}_{QS_j}^t)$ and $f_{2_j}^t(\overline{a}_{QS_j}^t)$, and $\tau_{1_j}^t(r_j)$ and $\tau_{2_j}^t(r_j)$ from the expressions reported in
Madsen et al. (2020).

2. Evaluate the exponential filters: $E_{1_j}^t = \exp\left(-\Delta t \frac{\overline{V}_{0_j}^t}{R}\frac{f_{1_j}^t}{\tau_{1_j}^t}\right)$ and $E_{2_j}^t = \exp\left(-\Delta t \frac{\overline{V}_{0_j}^t}{R}\frac{f_{2_j}^t}{\tau_{2_j}^t}\right)$.

3. Evaluate the DTU model correction:

$$\Delta u_j^t = A_1 Y_{1_j}^t + A_2 Y_{2_j}^t - \overline{u}_{QS_j}^t \qquad (A3)$$

with:

$$A_1 = 0.5847;\ A_2 = 0.4153; \qquad (A4)$$

$$Y_{1_j}^t = Y_{1_j}^{t-1} E_{1_j}^t + \overline{u}_{QS_j}^t\left(1 - E_{1_j}^t\right) \qquad (A5)$$

$$Y_{2_j}^t = Y_{2_j}^{t-1} E_{2_j}^t + \overline{u}_{QS_j}^t\left(1 - E_{2_j}^t\right) \qquad (A6)$$

Note that $Y_{1_j}^t$ and $Y_{2_j}^t$ correspond to the parameters $u_{i,y,1}^t$ and $u_{i,y,2}^t$ defined in Madsen et al. (2020). The names have been changed to avoid confusion with the notation.





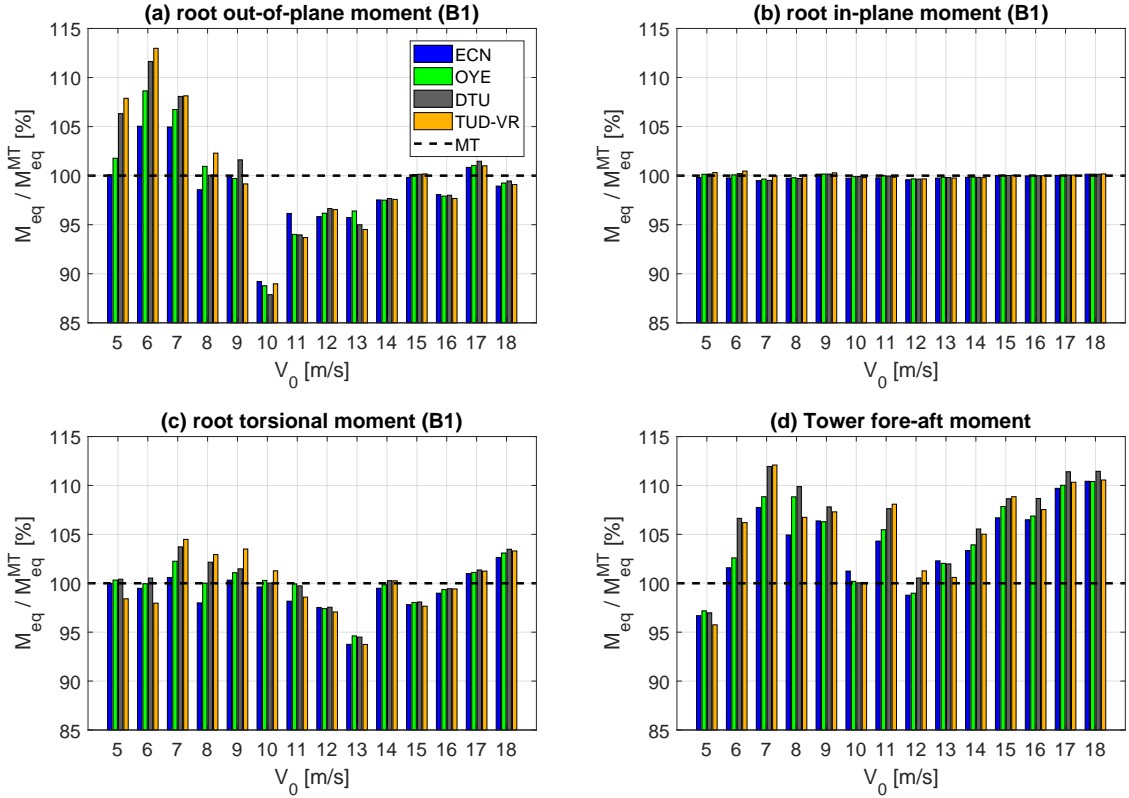

**Figure B1.** Here the results obtained with the original controller considering six seeds per wind speed (only the average value among the seeds is plotted) are reported in similar plots as those shown in fig. 10.

## Appendix B:  DLC 1.2 results with the official controller

Figure B1 shows the results of standard DLC 1.2 simulations featuring the official STRETCH turbine controller (including IPC). The plots are the same as in fig. 10, but the average loads among six seeds are shown for a wider set of wind speeds. For each wind speed, the same random seeds were used for the different dynamic inflow models. The results validity has been checked thoroughly, finding that the observed trends are well aligned with the standard deviations of both the operating quantities (i.e. blade pitch and rotor speed) and the loads. These differences are tightly linked to the controller behaviour. This makes their interpretation not only very difficult but also case specific, so the effect of each model on the equivalent loads might be hard to generalize from these results. The plot is anyhow left in this appendix for the interested reader to reflect upon.



*Author contributions.* SM implemented the models, run the simulations, and prepared the manuscript draft. KB and FS helped set up and run the aeroelastic simulations and interpret the results. GS provided expert guidance and project management support in all the steps of the

project. All the authors reviewed the manuscript and contributed actively to the development of the paper by providing continuous feedback and new ideas.

*Competing interests.* The authors declare that they have no conflict of interest.

*Acknowledgements.* This research has been supported by the Topsector Energy Subsidy from the Dutch Ministry of Economic Affairs and Climate (grant no. TEHE118020) with partners LM WindPower and GE Renewable Energy.





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
