# Peer review of "A comparison of dynamic inflow models for the Blade Element Momentum method"

_Wind Energy Science, 2022_

## Referee Comment (RC1)

A comparison of dynamic inflow models for the BEM

Simone mancini, koen boorsma, Gerard schepers, feike savenije

The paper compares the results from four dynamic inflow models for BEM (ECN, Oye, TUD-VR and DTU) and the lifting line model AWSM with a dynamic wake development. The ECN model Eq.(1) is very similar to the Pitt and Peters model and that could be mentioned. The right hand side is the total force from all blades and thus the resulting velocity, u, at the rotorplane is the same for all blades. This may not be true in case of e.g., a wind shear. Also, for completeness the force on the right hand should be divided by Prandtl's tip loss correction to correct for a finite number of blades. This is of course done in the code as described in the text but should also be reflected when stating the equation.

In line 130 is stated that tau1 is the slow time constant and tau2 the faster one in the Oye model Eqs.(2-3). It is the exact difference. In the Oye model the time constants are given as

$$\tau_1 = \frac{1.1R}{(1-1.3a^*)V_o}$$
$$a^* = \min(0.5, a)$$
$$\tau_2(r) = (0.39 - 0.26 \cdot (r/R)^2) \cdot \tau_1$$

And it is important to limit the axial induction factor in the denominator for tau1. Was this done in the BEM code used in this work ? Instead of solving the Oye differential equations discretely as shown in Appendix 1 one can alternatively solve the ODEs as shown in Hansen [1] to get them in a more filter like form using exponential functions as the DTU model Eq. (7) and that may be numerically more robust.

All the BEM based dynamic inflow models except the ECN model has a radial dependency term in the filtering functions. Both the DTU and TUD-VR models contain empirically determined simple functions found from actuator disc simulations when changing the disc load.

When applying a dynamic inflow model the results can depend very much on whether the induced velocities are calculated on the blades or on a stationary grid as described by Helge et al. [2]. In wind shear the loads will vary over one revolution. The loads are high when the blades are pointing straight up and low when the blades are pointing down, but because the time constants in the dynamic inflow models are large compared to the time for one revolution the induced wind becomes quite constant over one revolution. If, however, the induced winds are computed on a stationary grid and afterwards interpolated to the blades the induced wind will also be consistently high at the top positions and thus also have a shear and giving a different azimuthal angle of attack variation. The computations done in this paper is mostly for a constant wind speed so this effect may not be very visible but could play a role in the simulations made for DLC1.2, where there is a shear exponent of 0.14.

To avoid the influence of the controller in section 2.4.2 time series of pitch and rotational speeds are first made without any dynamic inflow model and used for all models. Is this a good idea, since the purpose of the dynamic stall models is to have dynamically more realistic inflow ? It is mentioned that this is probably the cause of higher equivalent loadings in the results shown in Fig. 10. When the controller is free to determine the pitch and rotor speed when applying the various dynamic stall models the equivalent loads are reduced after rated wind speed as shown in Fig. B1 in the appendix. Please comment more

It is very characteristic that the Oye model gives results close to the ECN model except near the tip and that the DTU and TUD-VR seems very similar, see Fig. 3 and 4 for the pitch step change. That these two last models give a similar dynamic response is probably simply because they are both empirically based on AD simulations to determine the time constants. In Fig. 3 showing the dynamic response of the induced wind at 40%, 60%, 80% and 95% for a step input of pitch angle it looks as if the AWSM code has almost no radial dependency. If this result is correct, then the radial dependency functions used in many of the BEM applied dynamic wake models are too strong. Please comment.

There seems to be some challenges with the dynamic airfoil response using the shed vorticity in the AWSM model when comparing to the BEM simulations using Theodorsen, but this is addressed in the paper.

[1] Aerodynamics of wind turbines 3[rd] ed, Earthscan, Hansen MOL

[2] Madsen HA et al., Implementation of the blade element momentum method on a polar grid and its aeroelastic load impact, wind energy science, 5(1), pp 1-27, 2020

---

## Referee Comment (RC2)

[referee-annotated manuscript omitted]

---

## Author Comment (AC1)

1 - The ECN model Eq.(1) is very similar to the Pitt and Peters model and that could be mentioned.
Thanks for the comment. A remark on their resemblance has been added in line 114.

2 - The right-hand side is the total force from all blades and thus the resulting velocity, u, at the rotorplane is the same for all blades. This may not be true in case of e.g., a wind shear. …
That is true indeed, but since this section only aims at introducing the dynamic inflow model it was preferred to avoid the implementation details of general (non-uniform) inflow conditions, and that is why axisymmetric conditions are mentioned in line 115.

… Also, for completeness the force on the right hand should be divided by Prandtl's tip loss correction to correct for a finite number of blades. This is of course done in the code as described in the text but should also be reflected when stating the equation.
In AM, the Prandtl's correction is not directly applied to the annulus force, but rather it is used to derive a local induced velocity from u (u_loc = u/F) that is then used to compute the local force Fx. This means that Fx accounts for the Prandtl's correction, but not with a simple multiplication. A brief remark on line 119 specifying that Fx includes Prandtl's correction has been added.
Should you have any further doubts about this, I would be more than happy to clarify them. However, a detailed description of all AM implementation details may not fit in the scope of this manuscript.

3 - In line 130 is stated that tau1 is the slow time constant and tau2 the faster one in the Oye model Eqs.(2-3). It is the exact difference. In the Oye model the time constants are given as:

$$\tau_1 = \frac{1.1R}{(1-1.3a^*)V_o}$$
$$a^* = \min(0.5, a)$$
$$\tau_2(r) = (0.39 - 0.26 \cdot (r/R)^2) \cdot \tau_1$$

Sorry, I cannot find the error that you are referring to. Tau2 is only a fraction of tau1 (i.e. $\tau_1 > \tau_2 > 0$), which makes it a smaller and thus faster time constant with respect to $\tau_1$. Also, $\tau_1$ only depends on the rotor loading and has no radial dependency, which makes it representative of the far wake effects. This looks fully consistent with previous works describing the model (e.g. Joule I & https://doi.org/10.5194/wes-2022-2). I hope this clarifies your doubt.

4 - It is important to limit the axial induction factor in the denominator for tau1. Was this done in the BEM code used in this work?
Thanks for checking. Yes, it was limited to 0.5 as prescribed in Joule I.

5 - Instead of solving the Oye differential equations discretely as shown in Appendix 1 one can alternatively solve the ODEs as shown in Hansen [1] to get them in a more filter like form using exponential functions as the DTU model Eq. (7) and that may be numerically more robust.
Thanks for the remark! I was not aware of this alternative implementation. A comment has been added in the appendix (line 510).

7 - When applying a dynamic inflow model the results can depend very much on whether the induced velocities are calculated on the blades or on a stationary grid as described by Helge et al. [2]. In wind shear the loads will vary over one revolution. The loads are high when the blades are pointing straight up and low when the blades are pointing down, but because the time constants in

the dynamic inflow models are large compared to the time for one revolution the induced wind becomes quite constant over one revolution. If, however, the induced winds are computed on a stationary grid and afterwards interpolated to the blades the induced wind will also be consistently high at the top positions and thus also have a shear and giving a different azimuthal angle of attack variation. The computations done in this paper is mostly for a constant wind speed so this effect may not be very visible but could play a role in the simulations made for DLC1.2, where there is a shear exponent of 0.14.

Thanks for the comment. Before commenting on the polar grid implementation, I would like to underline the fact that our annulus implementation of BEM is also capable of dealing with non-uniform wind (incl. shear) because the equations for each blade are solved separately resulting in a different annulus induced velocity value associated with each blade. To model dynamic wake effects, the dynamic inflow model is applied to the average of the annulus induction values obtained for the different blades, minimizing the influence of non-uniformities (which should not be filtered by the dynamic inflow model of course) on the computed correction. Below you can find a picture showing the azimuthal variation of the axial induced velocity at different spanwise locations for a steady case at 10m/s wind speed with a power law shear exponent of 0.5. The different lines compare the no dynamic inflow case (MT), with the DTU and TUD-VR dynamic inflow models. Being a non-uniform but steady case, no difference should be observed between the models. And that is almost the case except for some unwanted differences towards the blade tip (which require some further digging). The 1P variation of the induction (and thus loads) is therefore preserved.

With regards to the polar implementation proposed by Helge et al., which is mentioned in line 169 and in the paragraph starting at line 195, we really would like to investigate the differences with respect to our formulation, but this requires quite some restructuring of our code and there has not been enough time in this project. We anyhow keep a similar study high on our wish list for upcoming projects.

[Figure]

8 - To avoid the influence of the controller in section 2.4.2 time series of pitch and rotational speeds are first made without any dynamic inflow model and used for all models. Is this a good idea, since the purpose of the dynamic stall models is to have dynamically more realistic inflow? It is mentioned that this is probably the cause of higher equivalent loadings in the results shown in Fig.10. When the

controller is free to determine the pitch and rotor speed when applying the various dynamic stall models the equivalent loads are reduced after rated wind speed as shown in Fig. B1 in the appendix. Please comment more.

[Figure]

I am assuming that dynamic inflow models is meant in spite of dynamic stall. The equivalent load levels found with the different dynamic inflow models when the real controller (i.e. unconstrained) is used are in agreement with the differences in pitch and rotor speed variations (expressed with dimensions in plots -c- and -d-, and as a percentage of MT values in -e- and -f-). The equivalent load results are therefore consistent with what happens in the simulations, however, due to the tight interdependency of unsteady pitch, rpm, wind speed, and induction, it is quite difficult to understand what is causing what and thereby draw general conclusions on dynamic inflow model

effects. This is why it was decided to prescribe pitch and rpm as briefly mentioned in line 257 and in Appendix B.

The choice of the no-dynamic inflow (MT) pitch and rpm time histories as a reference is debatable. The intention was to use a "neutral" case that would make the comparison between the different DI models (which is the primary goal of the article) as fair as possible. Choosing a specific dynamic inflow model as a reference would also be possible, but it would be difficult to motivate why one was chosen instead of the others.

9 - It is very characteristic that the Oye model gives results close to the ECN model except near the tip and that the DTU and TUD-VR seems very similar, see Fig. 3 and 4 for the pitch step change. That these two last models give a similar dynamic response is probably simply because they are both empirically based on AD simulations to determine the time constants. In Fig. 3 showing the dynamic response of the induced wind at 40%, 60%, 80% and 95% for a step input of pitch angle it looks as if the AWSM code has almost no radial dependency. If this result is correct, then the radial dependency functions used in many of the BEM applied dynamic wake models are too strong. Please comment.

Thanks for the remark. The results have been double-checked once more to confirm the trends shown in the plots of Fig.3. The responses for the different radial locations are reported in the following plots to highlight the radial dependency of the normalized induction response for each model:

[Figure]

[Figure]

[Figure]

[Figure]

[Figure]

Note that, unlike ideal actuator disk cases, the radial variations are not only due to the radial dependency of the model parameters but also to the spanwise variation of aerodynamic loading. As a result, even if the dynamic inflow model parameters give a strong radial dependency the normalized responses of all two-constant models show limited variations with radius (as AWSM). These small variations do not have the same trend observed in AWSM though.

---

## Author Comment (AC2)

Dear reviewer,

Here you can find brief answers to your general comments and a few pictures supporting the detailed replies to all of your comments, which you may find in the appendix below.

A nomenclature is needed to understand all these terms, such as AM, AWSM, ECN, Oye,TUD-VR, etc.

Thank you for the suggestion. A nomenclature has been added before the introduction.

The quantified load of AWSM simulations should be included in the analysis of section 3.3 and Appendix B.

Running AWSM simulations with the same prescribed pitch and rpm controller used in BEM was not successful (except for the 14m/s simulation depicted below). However, as motivated in the detailed reply, adding AWSM results would not contribute much to the purpose of this paper because the differences that might be found could hardly be attributed to dynamic inflow modelling differences.

Please better justify the necessity of exploring AWSM without the shed vorticity in sections 3.1.1 and 3.2.1

Parts of the conclusions have been rephrased to make this more evident. Adding AWSM simulations with and without shed vorticity effects has allowed highlighting the different effects of the shed and trailed vorticity. This clearly confirmed several observations and hypotheses that were made in previous works (e.g. IEA Task 29 phase III). Although a bit artificial, we believe a similar distinction to be valuable when trying to understand how to improve the aerodynamic modelling of wind turbine rotors.
Here it is noted that a similar approach (i.e. comparing predictions with and without shed vorticity) is also being pursued in the IEA Task 47 to highlight the sources of difference in the aeroelastic predictions of different codes and models in a turbulent inflow case from the DANAERO experiment.

**Figures** (referring to the detailed replies in the appendix):

[Figure]

**Fig. 1:** AWSM induced velocity response example.

[Figure]

**Fig. 2:** same as figure 5 in the paper, but comparing Snel to the case without any dynamic stall model.

[Figure]

**Fig. 3:** same as figure 6 in the paper, but for the ABOVE-rated case. Due to the low (absolute) rotor loading above rated conditions, there is hardly any dynamic wake effect.

[Figure]

**Fig. 4:** same as figure 10 in the paper but including the AWSM simulation with prescribed pitch & rpm at 14m/s.

[revised manuscript text omitted]